# Learning by Directional Gradient Descent

**David Silver[1,2], Anirudh Goyal[3], Ivo Danihelka[1,2], Matteo Hessel[1], Hado van Hasselt[1,2]**

## Abstract

How should state be constructed from a sequence of observations, so as to best achieve some objective? Most deep learning methods update the parameters of the state representation by gradient descent. However, no prior method for computing the gradient is fully satisfactory, for example consuming too much memory, introducing too much variance, or adding too much bias. In this work, we propose a new learning algorithm that addresses these limitations. The basic idea is to update the parameters of the representation by using the directional derivative along a candidate direction, a quantity that may be computed online with the same computational cost as the representation itself. We consider several different choices of candidate direction, including random selection and approximations to the true gradient, and investigate their performance on several synthetic tasks.

## 1 Introduction

A key requirement of intelligence is the ability to integrate information over time, by constructing an internal representation of state from sequences of observations. This is critical for an agent to predict or control its future experience, whenever its environment is not fully observable (i.e. whenever individual observation are not sufficient for making the required predictions, or decisions). An efficient representation must construct state incrementally through an `update` function $s_{t+1} = \text{update}(s_t, o_t)$ with the previous state $s_t$ and the *last* observation $o_t$ as inputs. Since observations may include lots of redundant or irrelevant information, state representations must be tailored to a given task, encoding all and only the aspects of past experience required to perform such task.

To learn state representations from data, it is useful to be able to assess how changing the `update` function affects the long term performance of the agent. If the `update` function is parameterized by a differentiable function such as a recurrent neural network, we can measure the effect of changing each parameter in terms of a suitable loss function, and then update the parameter in the direction of steepest descent of the loss function, by using a suitable variant of (stochastic) gradient descent. Unfortunately, no existing method for computing the steepest descent direction of a recurrent function satisfies all the basic desiderata for a general approach to the problem. Prior methods either use too much memory (BPTT, Rumelhart et al. 1986; Werbos 1988), too much computation (RTRL, Williams & Zipser 1989), have too much variance (UORO, Tallec & Ollivier 2017; SPSA, Spall et al. 1992), or too much bias (DPG, Silver et al. 2014; synthetic gradients, Jaderberg et al. 2017b).

In this paper we propose an algorithm to efficiently perform near-steepest descent of recurrent functions, using the same computational and memory complexity as the forward computation of the recurrent function. The proposed approach is based upon the observation that the *directional* derivative of a recurrent function along any arbitrary direction $u$ can be computed efficiently and then can be used to construct a descent direction. When applied to recurrent networks, this observation allows the directional derivative of a recurrent function to be computed efficiently in a fully online manner, with computational cost the same as the forward computation of the function.

What direction $u$ is most effective? In the special case where $u$ matches the direction of the true gradient, we recover the steepest descent update. This suggests the following algorithm: first approximate the true gradient, and then follow a descent direction along the direction of the estimated gradient, in order to reduce the loss. We analyse and evaluate empirically random choices of direction, and also investigate directions based upon several methods of gradient approximation.

[01] DeepMind, London, UK, [2] University College London, [3] Mila, University of Montreal. Corresponding author: `anirudhgoyal9119@gmail.com`

## 2 METHODS FOR APPROXIMATING THE GRADIENT

In this section, we first give a brief overview of common methods for approximating true gradient.

***Notation.*** We consider a recurrent network represented by a differentiable function $(l_t, x_t) = f(x_{t-1}, o_t, w)$. The recurrent network maintains at each time-step $t$ an internal "hidden" state $x_t$ of size $n$, and receives an observation $o_t$. The recurrent function is parameterized by a vector $w$ of size $p$. At every time-step there is a loss $l_t$. For our purposes, any output of the network (such as its predictions or actions) may be considered to be part of the internal state $x_t$ that leads to a loss $l_t$. The objective is to minimise the total loss $L = \sum_{t=1}^{T} l_t$ over $T$ time-steps. A key requirement for efficient optimization is the computation of the gradient of the total loss with respect to the parameters, $\frac{\partial L}{\partial w} = \sum_{t=1}^{T} \frac{\partial l_t}{\partial w}$.

***Backpropagation-through-time (BPTT).*** Backpropagation-through-time (BPTT) uses backward differentiation to compute the gradient of the cumulative loss with respect to the parameters of a recurrent neural network. Specifically, the gradient can be computed by exchanging indices to consider the effect of state $x_t$ on all future losses $l_t, ..., l_T$:

$$\frac{\partial L}{\partial w} = \sum_{t=1}^{T} \frac{\partial l_t}{\partial w} = \sum_{t=1}^{T} \sum_{j=1}^{t} \frac{\partial l_t}{\partial x_j} \frac{\partial x_j}{\partial w} = \sum_{t=1}^{T} \left( \sum_{j=t}^{T} \frac{\partial l_j}{\partial x_t} \right) \frac{\partial x_t}{\partial w}. \tag{1}$$

The inner term may be accumulated by backpropagation of error,

$$\sum_{j=t}^{T} \frac{\partial l_j}{\partial x_t} = \frac{\partial l_t}{\partial x_t} + \sum_{j=t+1}^{T} \frac{\partial l_j}{\partial x_{t+1}} \frac{\partial x_{t+1}}{\partial x_t}.$$

This requires just $O(p)$ computation per time-step; however, the entire sequence must first be processed forward to compute the output. In practice, the memory-requirements of BPTT limit the computation of gradients to short windows, and limit the choice of neural network architecture to those with modest memory requirements. If the gradient of large neural networks must be computed over long horizons then BPTT may not be an appropriate choice of algorithm.

***Synthetic Gradients and Direct Policy Gradients.*** We now consider how one may use ideas drawn from reinforcement learning to estimate the future loss and hence allow for efficient online computation of gradients. One may estimate the loss using a value function $\tilde{v}(x_t) \approx \sum_{j=t}^{T} l_j$ and use the estimated loss in place of the true loss,

$$\frac{\partial L}{\partial w} = \sum_{t=1}^{T} \left( \sum_{j=t}^{T} \frac{\partial l_j}{\partial x_t} \right) \frac{\partial x_t}{\partial w} \approx \sum_{t=1}^{T} \frac{\partial \tilde{v}(x_t)}{\partial x_t} \frac{\partial x_t}{\partial w}, \tag{2}$$

where the loss may be estimated by bootstrapping from the next time-step, $\tilde{v}(x_t) \approx l_t + \sum_{j=t+1}^{T} l_j \approx l_t + \tilde{v}(x_{t+1})$. Alternatively, one may estimate the backpropagated gradient $\tilde{v}'(x_t) \approx \sum_{j=t}^{T} \frac{\partial l_{t+j}}{\partial x_t}$ and use the estimated gradient in place of the true gradient,

$$\frac{\partial L}{\partial w} \approx \sum_{t=1}^{T} \tilde{v}'(x_t) \frac{\partial x_t}{\partial w},$$

where the backpropagated gradient may be estimated by bootstrapping from the next time-step, $\tilde{v}'(x_t) \approx \frac{\partial l_t}{\partial x_t} + \sum_{j=t+1}^{T} \frac{\partial l_j}{\partial x_{t+1}} \frac{\partial x_{t+1}}{\partial x_t} \approx \frac{\partial l_t}{\partial x_t} + \tilde{v}'(x_{t+1}) \frac{\partial x_{t+1}}{\partial x_t}$.

It is worth noting that the former approach is equivalent to the Deterministic Policy Gradient (DPG) (Silver et al., 2014) in a reinforcement learning problem where the state is considered to be the input state $s_t = x_{t-1}$, the action is the next state $a_t = x_t$, the policy is the function mapping from state $x_{t-1}$ to action $x_t$ parameterized by weights $w$, and the reward is the negative loss $r_t = -l_t$ (Jaderberg et al., 2017b).

In practice, both deterministic gradients and synthetic gradients introduce significant bias into the estimate of the gradient. This has the downside that stochastic gradient descent, applied to the estimated gradient, may not actually descend the loss and may fail to converge.

***Real-time recurrent learning (RTRL).*** We now consider methods that compute the gradient of a recurrent function by forward differentiation. Given a function $z = h_T(...(h_1(y))...)$ with derivative $\frac{dz}{dy} = J_T...J_1$ where $J_t$ is the Jacobian of function $h_t$, one may instead compute the derivative $\frac{dz}{dy} = (J_T...(J_2(J_1)))$ by processing the chain rule in a forward direction and caching the intermediate quantity $\frac{dh_t}{dy} = J_t...J_1$. This is only computationally efficient when $y$ is smaller than $z$; this is the opposite case to usual in deep learning where we have many parameters for $y$ and a scalar loss for $z$.

Real-time recurrent learning (RTRL, Williams & Zipser, 1989) applies forward differentiation to compute the gradient of the cumulative loss with respect to the parameters of recurrent functions. This is achieved by caching the *sensitivity* of state to weights $\frac{dx_t}{dw}$, i.e. the total derivative of state to weights via all previous states,

$$\frac{\partial L}{\partial w} = \sum_{t=1}^{T} \frac{\partial l_t}{\partial w} = \sum_{t=1}^{T} \frac{\partial l_t}{\partial x_t} \frac{dx_t}{dw}.$$

The sensitivity may be accumulated via forward accumulation,

$$\frac{dx_{t+1}}{dw} = \frac{\partial x_{t+1}}{\partial w} + \frac{\partial x_{t+1}}{\partial x_t} \frac{dx_t}{dw}.$$

The size of the matrix $\frac{dx_t}{dw}$ is $n \times p$ and the computational cost of multiplying the Jacobian matrices in the second term is $O(pn^2)$, leading to a costly overall algorithm that is impractical for large-scale deep learning.

## 3 DEEP ONLINE DIRECTIONAL GRADIENT ESTIMATE (DODGE)

After explaining a directional derivative, we will introduce our online directional gradient estimate.

### 3.1 DIRECTIONAL DERIVATIVE

A basic property of gradients is that the *directional* derivative can be computed efficiently. One may efficiently compute the Jacobian-vector product $\frac{dz}{dy} \cdot u = (J_n...(J_2(J_1 \cdot u)))$ for any *tangent vector* $u$ of size $p$. A geometric interpretation of this product is the directional derivative of the gradient, i.e. $\frac{dz}{dy} \cdot u$ describes how much the gradient descends in the direction of the vector $u$.

### 3.2 GRADIENT ESTIMATE

When applied to recurrent networks, the directional derivative of a recurrent function can be computed efficiently with computational cost of just $O(p)$ per time-step, i.e. the same as the forward computation of the function.

$$\frac{\partial L}{\partial w} \cdot u = \sum_{t=1}^{T} \frac{\partial l_t}{\partial x_t} \left( \frac{dx_t}{dw} \cdot u \right)$$

$$\frac{dx_{t+1}}{dw} \cdot u = \frac{\partial x_{t+1}}{\partial w} \cdot u + \frac{\partial x_{t+1}}{\partial x_t} \left( \frac{dx_t}{dw} \cdot u \right)$$

For any unitary $u$, calculating the directional derivative along that direction provides a valid descent direction for updating weights, $\Delta w \propto \left( \frac{\partial L}{\partial w} \cdot u \right) u$.

The direction of steepest descent may be computed from many such tangent vectors, for example $\frac{\partial L}{\partial w} = \sum_{i=1}^{p} (\frac{\partial L}{\partial w} \cdot e_i) e_i$ for the unit bases $e_1, ..., e_p$. This requires $p$ separate directional derivatives. Alternatively, random directions such as $u_i \sim \{-1, +1\}^p$ may be sampled, leading to an accumulated gradient estimate $\frac{\partial L}{\partial w} \approx \frac{1}{n} \sum_{i=1}^{n} (\frac{\partial L}{\partial w} \cdot u_i) u_i$, but at the cost of introducing significant variance. However, if the tangent vector is chosen to be the gradient direction itself, $u = \frac{g}{\|g\|}$ where $g = \frac{\partial L}{\partial w}$, then only one such directional derivative is required,

$$(g \cdot u)u = \left( g \cdot \frac{g}{\|g\|} \right) \frac{g}{\|g\|} = g.$$

We are often interested in the expectation of the gradient, $\mathbb{E}[g]$ rather than the sample gradient. For example, stochastic gradient descent (SGD) averages over sample gradients to descend in the direction of the overall gradient. When using directional gradient descent, one may either compute the directional derivative of an individual sample gradient, and then average, or compute the directional derivative in the direction of the expected gradient $u = \frac{\bar{g}}{\|\bar{g}\|}$ where $\bar{g} = \mathbb{E}[g]$, and then average:

$$\mathbb{E}[(g \cdot u)u] = \mathbb{E}\left[\left(g \cdot \frac{\bar{g}}{\|\bar{g}\|}\right)\frac{\bar{g}}{\|\bar{g}\|}\right] = \left(\bar{g} \cdot \frac{\bar{g}}{\|\bar{g}\|}\right)\frac{\bar{g}}{\|\bar{g}\|} = \bar{g}.$$

This suggests a novel strategy of estimating the expected gradient $\bar{g}$ and using the estimated gradient as a descent direction $u$ in $(\frac{\partial L}{\partial w} \cdot u)u$. Note that, regardless of the quality of the gradient estimate, the algorithm will always follow a valid descent direction. A valid descent direction has a non-negative dot-product with the true gradient. The quality of the gradient solely determines the steepness of descent in expectation. In the next section, we consider methods that provide such an estimate of the gradient. We refer to this family of algorithms as *Deep Online Directional Gradient Estimate (DODGE)* and they can be understood as *near*-steepest descent algorithms.

```python
def dodge(loss_function, params, direction):
    # Compute the Jacobian-vector product.
    loss, loss_jvp = jax.jvp(loss_function, [params], [direction])
    grad_estimate = loss_jvp * direction
    return loss, grad_estimate
```

Listing 1: DODGE implemented in JAX.

### 3.3 METHODS FOR DODGE

We now consider several possible methods for estimating the expected gradient $\bar{g}$. When training on data from a stationary distribution, $\bar{g}$ may be estimated by averaging over many approximate sample gradients $\bar{g} \approx \mathbb{E}[\tilde{g}]$. In the proposed update rule, the DODGE algorithm updates the gradient estimate using a moving average $\Delta\bar{g} = \alpha(\tilde{g} - \bar{g})$ and update the weights using SGD, $\Delta w = -\beta(\frac{\partial L}{\partial w} \cdot \frac{\bar{g}}{\|\bar{g}\|})\frac{\bar{g}}{\|\bar{g}\|}$. The sample gradient $\tilde{g} \approx \frac{\partial L}{\partial w}$ may be approximated in many ways. We consider several methods:

**ORACLE** As a baseline, the exact sample gradient $\tilde{g} = \frac{\partial L}{\partial w}$.

**TBPTT** Truncated Backpropagation Through Time can propose a biased gradient $\tilde{g}$. DODGE is able to convert the biased gradient to a valid descent direction.

**DPG** The deterministic policy gradient $\tilde{g} = \sum_{t=1}^{T} \frac{\partial \tilde{v}(x_t)}{\partial x_t}\frac{\partial x_t}{\partial w} \approx \frac{\partial L}{\partial w}$.

**SYN** The synthetic gradient $\tilde{g} = \sum_{t=1}^{T} \tilde{v}'(x_t)\frac{\partial x_t}{\partial w} \approx \frac{\partial L}{\partial w}$.

**Computational cost:** The computational cost of the proposed method is double than that of the forward computation of the function i.e., one forward pass for computing the candidate direction, and another forward pass for computing the directional derivative.

## 4 DODGE MATHEMATICAL PROPERTIES

Given a direction $u$, DODGE estimates the $i$th element of the gradient $g$ by

$$\hat{g}(i) = u(i)\sum_k u(k)g(k). \tag{3}$$

We will list several properties of this estimate, with more properties listed in Appendix A.

### 4.1 NON-NEGATIVE DOT-PRODUCT.

DODGE has non-negative dot-product with the gradient:

$$\hat{g} \cdot g = ((g \cdot u)u) \cdot g = (g \cdot u)^2 \geq 0. \tag{4}$$

The non-negative dot-product enables that the expected loss is lower after an update with a small enough step size (Bottou et al., 2018).

## 4.2 RANDOM DIRECTIONS

***Unbiased estimate with random directions.*** If the $u$ elements are independent with zero mean and unit variance, then DODGE is an unbiased estimate of the gradient:

$$\mathbb{E}\left[\hat{g}(i)\right] = \mathbb{E}\left[u(i)\sum_k u(k)g(k)\right] = \mathbb{E}\left[g(i) + u(i)\sum_{k\neq i} u(k)g(k)\right] = \mathbb{E}\left[g(i)\right]. \tag{5}$$

***Variance with random signs.*** If each $u$ element contains a random sign from $\{-1, 1\}$, then the variance of $\hat{g}(i)$ is

$$\mathbb{E}\left[(\hat{g}(i) - \mathbb{E}\left[g(i)\right])^2\right] = \mathbb{E}\left[\hat{g}^2(i)\right] - \mathbb{E}^2\left[g(i)\right] = \mathbb{E}\left[\left(\sum_k u(k)g(k)\right)^2\right] - \mathbb{E}^2\left[g(i)\right] \tag{6}$$

$$= \sum_k \mathbb{E}\left[g^2(k)\right] - \mathbb{E}^2\left[g(i)\right] \tag{7}$$

$$= \mathbb{E}\left[(g(i) - \mathbb{E}\left[g(i)\right])^2\right] + \sum_{k\neq i} \mathbb{E}\left[g^2(k)\right], \tag{8}$$

where we used the fact that the variance of a sum of independent terms equals to a sum of variances. Notably, the variance is proportional to the number of parameters and not affected by the size of the state representation.

## 4.3 MULTIPLE RANDOM DIRECTIONS

There are two simple approaches to use multiple random directions.

***Average of estimates.*** The first approach would use multiple independent random directions and average the obtained estimates of the gradient. If using $N_{\text{dirs}}$ directions with random signs and if each $\mathbb{E}\left[g^2(k)\right]$ is $\sigma^2$, the average of multiple estimates has variance

$$\mathbb{E}\left[(\hat{g}_{\text{avg}}(i) - \mathbb{E}\left[g(i)\right])^2\right] = \mathbb{E}\left[(g(i) - \mathbb{E}\left[g(i)\right])^2\right] + \frac{p-1}{N_{\text{dirs}}}\sigma^2, \tag{9}$$

where $p$ is the number of parameters.

***Direction partitioning.*** Another approach would be to generate one random direction and partition it to multiple parts, padded with zeros. The gradient estimate is then obtained by the sum of estimates from the parts. This approach has lower variance as compared to using the average of the estimates.

If using $N_{\text{dirs}}$ parts, each with $\frac{p}{N_{\text{dirs}}}$ random signs, this approach has variance

$$\mathbb{E}\left[(\hat{g}_{\text{partitioned}}(i) - \mathbb{E}\left[g(i)\right])^2\right] = \mathbb{E}\left[(g(i) - \mathbb{E}\left[g(i)\right])^2\right] + \frac{p - N_{\text{dirs}}}{N_{\text{dirs}}}\sigma^2. \tag{10}$$

If the number of directions $N_{\text{dirs}}$ is equal to the number of parameters $p$, each $g(i)$ can be estimated perfectly. In this extreme, each direction is a one-hot vector with a random sign. We also note that this case can be seen as a generalisation of RTRL. In the case of the number of directions equal to the number of parameters (i.e., $N_{\text{dirs}} = p$), DODGE would be mathematically equivalent to RTRL.

## 4.4 GEOMETRIC INTERPRETATION OF DODGE

Consider a set of $k$ orthonormal directions $U = [u_1, ..., u_k]^\top$. For example, in high-dimensional spaces, independent random directions are near-orthogonal with high probability. We project the gradient onto the subspace spanned by those directions, $\frac{\partial L}{\partial w}P_U = \frac{\partial L}{\partial w}UU^T = \sum_{i=1}^k \left(\frac{\partial L}{\partial w} \cdot u_i\right) u_i$. This projection is computed online by DODGE from $k$ directional derivatives $u_i \cdot \frac{\partial L}{\partial w}$, at a cost of $k$ times the forward computation. The projected gradient may be used to update parameters, e.g. $\Delta w = \alpha \frac{\partial L}{\partial w}UU^T$, which provides a descent direction, $L(w + \Delta w) \leq L(w)$ for sufficiently small $\alpha$, with equality only at local minima or when the gradient is orthogonal to the subspace.

## 5    EXPERIMENTS

We now report and discuss the results of an empirical study that analyses the performance of the proposed estimator using different tasks, as well as using different ways to approximate the expected gradient. We use JAX (Bradbury et al., 2018) to implement all experiments. To implement DODGE, we need to compute a Jacobian-vector product, and JAX has an excellent support for Jacobian-vector products. We provide an example DODGE implementation in Listing 1.

### 5.1    PROBLEMS

We evaluate the proposed DODGE update on different problems. We first give a brief description of the different problems and then in the next section we discuss different ways of getting the direction $u$ required to compute the DODGE update. We now describe the details of our implementation. On sequence modeling tasks, we use an LSTM network (Hochreiter & Schmidhuber, 1997) with 128 units and a batch size of 32. We optimize the log-likelihood using the Adam optimizer (Kingma & Ba, 2014). For each method, we choose the best learning rate from $\{0.003, 0.001, 0.0003, 0.0001, 0.00003, 0.00001\}$, based on the final performance. We repeat each experiment 5 times with 5 different random seeds.

***Copy task.*** The copy problem defined in Graves et al. (2014) is a synthetic task designed to test the model's ability to remember a sub-sequence for a large number of time-steps. Here, we would like to test out whether near-steepest descent can help to learn long-term dependencies.

***MNIST classification task.*** It is a database of handwritten digits (LeCun, 1998) that is commonly used for training and evaluating image classification models.

***Influence Balancing task.*** This task was introduced by Tallec & Ollivier (2017) to study the bias in the gradient estimate from truncated backpropagation through time. This task is particularly sensitive to short-horizon bias, as the gradient for the single parameter $\theta \in R$ has the wrong sign when estimated from short unrolls.

***Image regression NeRF task.*** This task trains the initial parameters of a 2D-NeRF model (Mildenhall et al., 2020), represented by an MLP. The network takes in a pixel coordinate $(x, y)$, and then outputs the $(R, G, B)$ color value at the particular position. The learned initial parameters need to support fast adaptation for each new image. We build upon the experimental setup proposed by Tancik et al. (2021).

### 5.2    SANITY CHECK

We empirically show that for a model with a single parameter, the DODGE update is equivalent to RTRL. In order to explore this we use the influence balancing task. This task involves learning of a scalar parameter. We use the same experiment setup as in (Tallec & Ollivier, 2017). Since we are training only a single parameter, we use a random direction. In Table 1, we empirically show that the implementation of the proposed method achieves exactly the same performance as the exact RTRL.

Table 1: **Influence balancing task**: TBPTT with a short truncation length diverges, while DODGE with a random direction performs exactly the same as RTRL.

| Method | Loss |
|---|---|
| TBPTT (truncation length 100) | 0.93 |
| RTRL | 1.11e-12 |
| DODGE | 1.11e-12 |

### 5.3    RANDOM DIRECTIONS AS CANDIDATE DIRECTIONS

***Increasing the number of random directions helps.*** We first show that a random direction can be converted to a valid descent direction. In order to test this, we train the model using a DODGE update on the MNIST classification task. Figure 1 shows the result of training a model by DODGE with random directions. It also shows that increasing the number of random directions improves the learning speed, but learning is possible and stable even with a single random direction. We also test the use of the random directions on the copy task. Figure 4b shows DODGE on the copy task. There, DODGE with random directions achieves low loss only when using 64 random directions, which may not be feasible for networks with many parameters.

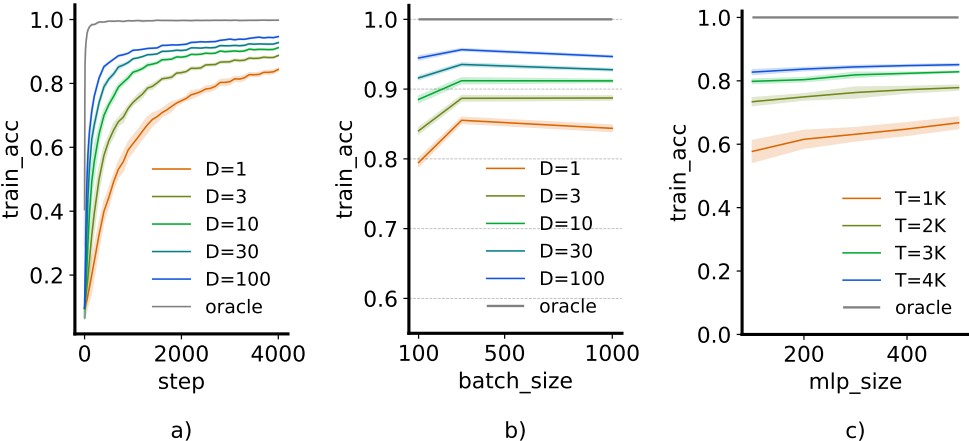

Figure 1: Training accuracy on MNIST for feedforward neural networks trained by DODGE with random directions. **a)** We report the accuracy as a function of the number of updates. In grey, DODGE with oracle steepest descent directions. From orange to dark blue, DODGE with increasing numbers of directions sampled on each update, from 1 to 100. Estimating the gradient using more directions improves performance, but learning is possible and stable even with a single direction. **b)** A parameter study of the joint effect of the number of directions and the batch size used to evaluate the directional derivative along each direction. We evaluate the accuracy after seeing a fixed number of samples (regardless of how they are batched). Increasing batch size from 100 to 300 improves performance, especially when using a single direction $u$ (in orange). Increasing the batch size further (to 1000) however decreases performance. **c)** A parameter study of the effectiveness of DODGE for different network sizes. The accuracy is reported as a function of the number of units in the two hidden layers of the multi-layer perceptron used in the experiment. The different lines correspond to different snapshots during training: after 1000, 2000, 3000, 4000 updates, respectively.

## 5.4 APPROXIMATE CANDIDATE DIRECTIONS: TBPTT, REPTILE, AND SYNTHETIC GRADIENTS

As discussed in Section 3.3, there are various ways to approximate the true gradient. For example, one can approximate the true gradient either via the use of truncated backprop, which reduces memory consumption at the cost of increasing bias, or the use of first-order algorithms for meta-learning that do not differentiate through the inner loop, or by learning a critic that estimate the true loss and its gradient. Here we explore using these different methods for estimating the expected gradient, and then using the estimated gradient as a candidate direction for computing the directional derivative.

### 5.4.1 DIRECTION AS A RESULT OF TRUNCATED BACKPROPAGATION

In order to evaluate whether the DODGE update can help in learning long term dependencies we evaluate DODGE on the copy task. For all our experiments, we use a sequence of length 25. Truncated backpropagration through time (TBPTT) for training recurrent networks is known to have trouble remembering inputs it has seen many steps ago. In order to get the direction used for calculating the DODGE update, we randomize the truncation length across different batches by uniformly sampling the truncation length from $\{1, \ldots, \texttt{max\_trunc\_len}\}$.Tallec & Ollivier (2017) used importance sampling to unbias the truncated gradients and required a longer maximum truncation length. Figure 2 shows the results of using a direction from TBPTT to compute a DODGE update. The four different plots are for different truncation lengths. The DODGE update helps mainly on short truncation lengths ($\texttt{max\_trunc\_len} = 3$).

### 5.4.2 DIRECTION AS A RESULT OF REPTILE.

Meta-learning seeks to learn a network that can adapt quickly on a previously unseen task. Model-Agnostic Meta-Learning (MAML, Finn et al., 2017) uses an outer loop of gradient based learning to learn highly adaptive initial parameters. The algorithm unrolls in the inner loop and differentiates through the resulting computation graph. Alternatively, Reptile (Nichol et al., 2018) updates the initial parameters towards the final parameters learned on a task. Because Reptile does not backpropagate through the inner loop of the computational graph, Reptile saves memory at the cost of obtaining biased gradient estimates.

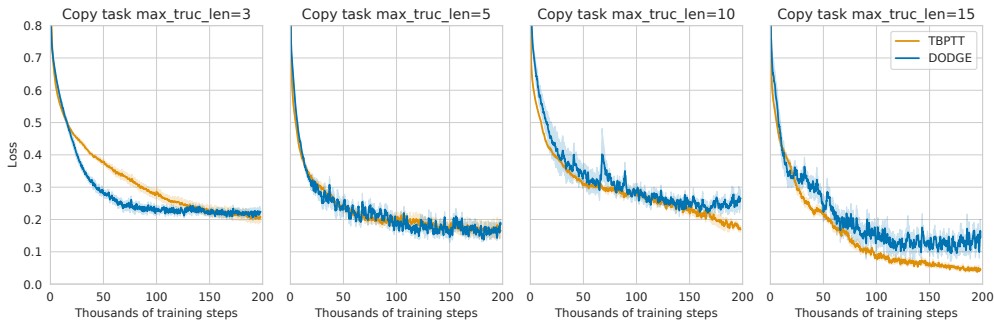

Figure 2: DODGE with a direction from truncated backpropagation through time (TBPTT). We plot cross-entropy (Y axis) as a function of the training steps (X axis). Lower cross-entropy is better. Shades denote standard errors from 5 seeds.

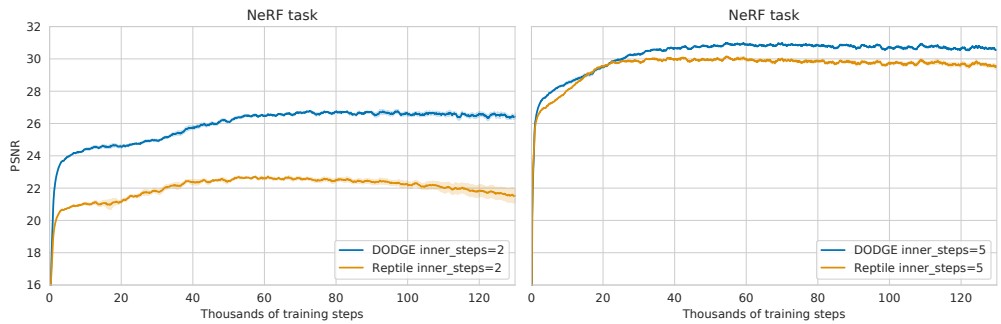

Figure 3: **Direction from the update proposed by Reptile.** DODGE as a result of using a direction proposed by Reptile. We plot Peak Signal-to-Noise Ratio (PSNR; Y axis) as a function of the training-steps (X axis). Higher PSNR is better. PSNR is commonly used to quantify the reconstruction quality for images.

Figure 3 shows the experimental results on the image regression NeRF task with CelebA images (Liu et al., 2015). For DODGE, we use as the direction the update proposed by Reptile. The figure shows that the network optimized by DODGE learns faster, especially in the setting with two inner steps.

### 5.4.3 DIRECTION AS A RESULT OF LEARNING A CRITIC: SYNTHETIC GRADIENT.

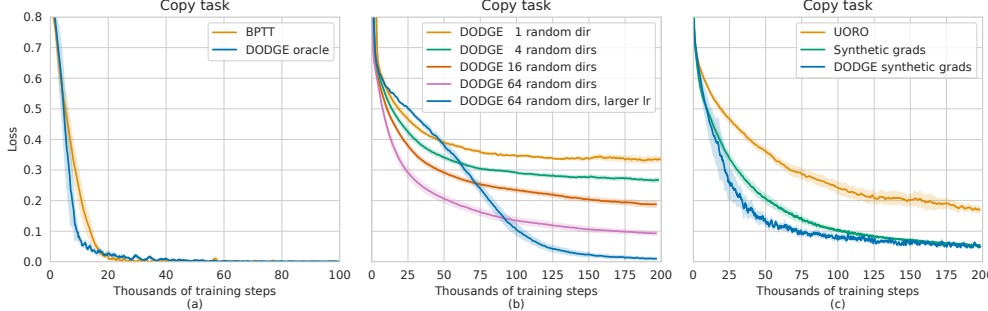

Figure 4: We plot cross-entropy (Y axis) as a function of training-steps (X axis) on the copy task. Lower cross-entropy is better. **(a)** DODGE using an oracle direction (i.e., the gradient computed by BPTT on the previous minibatch). **(b)** DODGE using multiple random directions. Multiple random directions reduce the variance and allow to use a larger learning rate. **(c)** DODGE using a direction proposed by a critic (learned using synthetic gradients).

We now consider ideas from reinforcement learning which allows one to estimate the future loss by learning a value function and then using the estimated loss in place of the true loss. We use synthetic

gradients for learning the critic as proposed by Jaderberg et al. (2017a). For analyzing the DODGE update using synthetic gradients, we use the copy task of length 25 as in the case of TBPTT.

Figure 4c shows the cross entropy values plotted against the number of training steps using the DODGE update compared to the synthetic gradient baseline. Here also the network trained with the DODGE update optimizes faster as compared to the baseline.

## 6 RELATED WORK

Online credit assignment in RNNs without backtracking remains an open research problem. RTRL (Williams & Zipser, 1989) attempts to solve this problem by estimating gradients using a forward mode automatic differentiation instead of backpropagation. Forward mode automatic differentiation allows for computing unbiased gradient estimates in an online fashion, however, it normally requires storage of the gradient of the current hidden state values with respect to the parameters, which is $O(N^3)$ where $N$ is the number of hidden units. The Unbiased Online Recurrent Optimization (UORO) (Tallec & Ollivier, 2017) method gets around this by updating a rank-1 approximation, which is shown to keep the estimate of the gradient unbiased, but at the cost of increasing the variance of the gradient estimator. DODGE with random signs can also benefit from the variance reduction method proposed in UORO, by rescaling the random signs (if we know the variance of the gradient elements). Various approaches use sparse approximations for real time recurrent learning, like SnAp (Menick et al., 2020), which provides a deterministic biased gradient and can be combined with DODGE (by using SnAp to provide a candidate direction). Optimal Kronecker-Sum Approximation (OK, Benzing et al., 2019) is designed for a specific network architecture and achieves low variance at a higher computation cost than UORO. Interestingly, DODGE with a random direction can be seen as an unexplored special case of the generalized UORO estimator by Cooijmans & Martens (2019). More recently, Vicol et al. (2021) introduced Persistent Evolution Strategies (PES), a method for unbiased gradient estimation by estimating gradients from truncated unrolls and allowing frequent parameter updates. On the influence balancing task, DODGE with one random direction reproduces RTRL, while PES requires $10^3$ particles.

Previously, directional derivatives were successfully used for Bayesian optimization with a random direction (Ahmed et al., 2016) or with a direction based on the value of information (Wu et al., 2017). The proposed estimator is also related to the "sketch and project" gradient estimate (Hanzely et al., 2018), which uses a sketch to construct an unbiased estimator of the gradient. With DODGE, we consider adaptive estimates of the true gradient to provide a candidate direction (i.e., to construct the sketch). To the best of our knowledge, these methods have not been studied specifically in the context of dynamic models (i.e., training of the recurrent architectures).

## 7 CONCLUSION, LIMITATIONS AND FUTURE WORK.

***Conclusion.*** In this work, we proposed the idea of using a directional derivative to compute the descent direction for updating the parameters of a neural network. We analyzed different ways to get candidate directions, including random selection and estimating directions that may be close to the true gradient. The proposed learning algorithm computes the directional derivative along a descent direction which is then used to update parameters. We empirically showed that the DODGE update helps to improve the convergence when gradient is approximated through different choices of candidate direction. We also showed that multiple random directions can be combined to reduce the variance.

***Limitations and Future Work.*** Here, we focus on building a strong foundation by carefully analyzing the proposed estimator in various different scenarios (i.e., different approximations to true gradient). A limitation of the current study is that we don't evaluate on problems where the credit needs to be assigned to states or actions which may be separated by thousands of time-steps like often the case for problems in reinforcement learning. Another limitation is that for the problems involving approximate candidate directions we limit our simulations to the scenario where we only use a single direction as compared to either using multiple directions at a particular time-step or by changing the directions in an online fashion. Our experiments with random directions show that using multiple random directions improves the convergence. Future work would investigate using multiple approximate candidate directions for updating parameters.

ACKNOWLEDGMENTS

We would like to thank Charles Blundell and Fabio Viola for helpful advice and help with experiments.

ETHICS STATEMENT

The authors do not foresee any negative social impacts of this work, but of course the accumulation of improvements in ML could be misused as it may give more power to nefarious agents.

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

## A    ADDITIONAL DODGE MATHEMATICAL PROPERTIES

Given a direction $u$, DODGE provides the following estimate of the gradient:

$$\hat{g} = (g \cdot u)u. \tag{11}$$

We will list more properties of this estimate.

**A direction can recover the expected Adam's update.** We need to be careful when using DODGE with an advanced optimizer. Many optimizers assume that the gradient estimate is unbiased. For example, if combining non-random DODGE with Adam (Kingma & Ba, 2014), we would not have a non-negative dot-product of the Adam's update with the expected gradient. The product of Adam's diagonal matrix and DODGE's $uu^T$ matrix is not always a positive definite matrix. We can instead incorporate Adam's scaling inside the direction $u$.

Given a scale $c(i) > 0$ for each gradient element, we want to obtain $c(i)\mathbb{E}\left[g(i)\right]$. If $u(i) = \frac{c(i)\mathbb{E}[g(i)]}{\sqrt{\sum_k c(k)\mathbb{E}^2[g(k)]}}$, the expected estimate becomes the wanted quantity:

$$\mathbb{E}\left[\hat{g}(i)\right] = \mathbb{E}\left[u(i)\sum_k u(k)g(k)\right] = c(i)\mathbb{E}\left[g(i)\right]. \tag{12}$$

### A.1    RANDOM DIRECTIONS WITH A BASELINE

**Unbiased estimate with a baseline.** We will mention the possibility to combine DODGE with a baseline. Let us define DODGE with a baseline as

$$\hat{g}_b(i) = u(i)\sum_k u(k)g(k) - u(i)\sum_{k \neq i} u(k)b(k), \tag{13}$$

where $b(k)$ is the $k$-th element of the baseline $b \in \mathbb{R}^p$. If he $u$ elements are independent with zero mean and unit variance and if $b(k)$ is independent of $u(k)$, then $u(k)b(k)$ has zero mean and $\hat{g}(i)$ is an unbiased estimate of $g(i)$.

**Variance with random signs and a baseline.** If $b$ is independent of $u$ and if each $u$ element contains a random sign from $\{-1, 1\}$, then the variance of $\hat{g}_b(i)$ is

$$\mathbb{E}\left[(\hat{g}_b(i) - \mathbb{E}\left[g(i)\right])^2\right] = \mathbb{E}\left[\hat{g}_b^2(i)\right] - \mathbb{E}^2\left[g(i)\right] \tag{14}$$

$$= \mathbb{E}\left[\left(g(i) + u(i)\sum_{k \neq i} u(k)(g(k) - b(k))\right)^2\right] - \mathbb{E}^2\left[g(i)\right] \tag{15}$$

$$= \mathbb{E}\left[(g(i) - \mathbb{E}\left[g(i)\right])^2\right] + \sum_{k \neq i} \mathbb{E}\left[(g(k) - b(k))^2\right]. \tag{16}$$

The $g$ would be the perfect baseline. Unfortunately, the perfect baseline is hard to obtain.

## A.2    HYPER-PARAMETERS.

**Nerf task.**    We build upon the experimental setup as proposed by Tancik et al. (2021).

**Copy Problem.**    On sequence modeling tasks, we use an LSTM network (Hochreiter & Schmidhuber, 1997) with 128 units and a batch size of 32. We optimize the log-likelihood using the Adam optimizer (Kingma & Ba, 2014). For each method, we choose the best learning rate from $\{0.003, 0.001, 0.0003, 0.0001, 0.00003, 0.00001\}$, based on the final performance. We repeat each experiment 5 times with 5 different random seeds.

For experiments using direction as a result of learning a critic (synthetic gradient), we train for 200K gradient steps with a batch size of 32. To train the critic we use truncation length of 5, and syngrad scale of 0.5, and we use quadratic critic as recommended in (Jaderberg et al., 2017b).

