# OpenReview forum: "Learning by Directional Gradient Descent"
_ICLR.cc/2022/Conference — ICLR 2022 Poster_

### Official Review · Reviewer_AjYD · 2021-11-01

**Correctness:** 2
**Technical Novelty And Significance:** 1
**Empirical Novelty And Significance:** Not applicable
**Recommendation:** 1
**Confidence:** 5

**Main Review:**

The paper is not written well. And the mathematics is not clear. I recommend rejecting this paper.

**Summary Of The Paper:**

The paper applies the direction derivative to compute the learning problems.

**Summary Of The Review:**

The paper discusses the possibility of applying direction derivatives to improve the gradient descent method. There is no concrete calculations and analytical examples to support the idea.

---

### Official Review · Reviewer_brku · 2021-11-03

**Correctness:** 4
**Technical Novelty And Significance:** 3
**Empirical Novelty And Significance:** 3
**Recommendation:** 6
**Confidence:** 3

**Main Review:**

# Preliminary remarks

The proposed method is surprisingly simple. But it seems that, after a quick search, nothing close has been proposed before. I did not find any reference to methods mixing "directional gradient/derivative", "Monte Carlo estimation of the gradient", etc.

So, despite its simplicity, "DODGE" seems to be quite new. However, I might have missed some references.

Reference about "random coordinate descent", close to the main idea of the paper : *Langevin Monte Carlo: random coordinate descent and variance reduction*, Ding 2021.

# Strengths

The simplicity of the method is one of its main strengths. The idea of projecting the true gradient on an estimated gradient is the main practical application of the paper. The author show is one specific case that DODGE + synth. grads is better than synth. grands alone (Fig. 4).

# Weaknesses

The range of experiments is very narrow. The reader may expect more datasets for the same setup, in order to show that the results are consistent over *several* tasks and datasets (and not only one per setup).

# Clarity

Overall, the paper is easy to follow. However:
 * there are many typos (see below);
 * the "Related work" section may be placed just after the introduction. Since the proposed technique is surprisingly simple, the reader would expect a quick review of related papers very early in the article (and not at the end);
 * I would not recommend the use of a Jax code excerpt on page 4 (the reader is supposed to know how it works). Pseudo-code would be preferable, and this except could be put in the Appendix.

# Typos

* p 3: "for any $u$" => "for any unitary $u$"
* p 4: in the code excerpt: "gradient_estimate" becomes "grad_estimate"
* p 4: "ORACLE": bad alignment
* end of p 5: "<=" => "$\leq$"
* p 6, Sec. 5.2: "RTRL": bad alignment
* p 8: fig. 2: titles: "max_truc_len" => "max_trunc_len"

**Summary Of The Paper:**

This paper proposes to adapt the RTRL technique when training RNNs to make it usable in practice. That is, instead of computing the full gradient of the loss $L$ according to all parameters, the proposed method "DODGE" consists in computing the gradient of $L$ in *one* or a small number of directions in the parameter space. These "directional gradients" of $L$ are much easier to compute than the full gradient when using RTRL. More precisely, the user can choose a subspace $\mathcal{P}'$ (of dim. $p'$) of the space of parameters $\mathcal{P}$ (of dim. $p$), and compute the projection of the true gradient $g^*$ of $L$ on the space $\mathcal{P}'$, *with a computational cost proportional to $p'$*. This is advantageous when $p' \ll p$.

Notably, this method can be used in two ways:
 * estimating the true gradient by computing its projection on a random low-dimensional subspace of the parameters;
 * improve an existing estimation $\hat{g}$ of the true gradient $g^*$ by computing the projection of $g^*$ on $\hat{g}$: this projection has better properties than $\hat{g}$.

**Summary Of The Review:**

The idea is new, simple, relatively easy to implement, and seems to improve the estimation of the gradient. But the paper lacks empirical evidence, in particular consistency of the method over several datasets *with the same setup*.

---

> ### Author Response · Authors · 2021-11-18
> **Clarity + different candidate directions.**
>
>
> We thank the reviewer for taking time in providing feedback to our work. We are enthused that the reviewer found the idea “simple”.
>
> **Clarity and Typos**
>
> We thank the reviewer for taking time and carefully pointing out several typos. We would incorporate them in the next revision of our work. We would move the related work section after the introduction.
>
> **The range of experiments is very narrow.**
>
> As other reviewers point out, we evaluate the proposed idea on different tasks, and compared to various different baselines. In our experiments, we show that the candidate direction can come from various sources such as randomly sampled directions, truncated back-propagation through time (BPTT), the “Reptile” meta-learning approach by Nichol et al. (2018), or synthetic gradients.
>
> We evaluate the idea using these different candidate directions on different tasks such as copy task, NeRF task and the MNIST task. On the MNIST task, we also showed that the performance of the proposed method can be improved on using multiple candidate directions.
>
> Are their any particular experiments which the reviewer would like to see ?

---

> > ### Comment · Reviewer_brku · 2021-11-30
> > **Update**
> >
> > I have read the other reviews, and I disagree with reviewer GYWv about the novelty of the proposed method. It is true that, formally, the proposed method is *close to* (and not identical to) "sketch and project", but the aim is completely different. The "sketch and project" method is a variance reduction technique, the proposed method is a way to reduce the computational cost in a recurrent setting.
> >
> > Since the proposed method has been specifically designed as an improvement of RTRL for RNNs, it is more a variation of RTRL than a variation of "sketch and project".
> >
> > The additions about the computational cost are useful, and the were actually missing in the first version of the paper.
> >
> > However, I am still disappointed by the small number of experiments. For instance, it would have been interesting to plot Figure 4 for several common tasks, such as word or character prediction. Otherwise, the method is difficult to evaluate.
> >
> > As such, I do not change my recommendation. I did not see any flaw in the theory and the idea seems to be useful, but the experiments are on the border.

---

> > > ### Author Response · Authors · 2021-11-30
> > > **Thanks for your reply.**
> > >
> > > We thank the reviewer for engaging in discussion, and providing useful feedback. We did another experiment to address reviewer's concerns.
> > >
> > > **The sketch and project method is a variance reduction technique, the proposed method is a way to reduce the computational cost in a recurrent setting**
> > >
> > > We thank the reviewer for providing their perspective.
> > >
> > > **I am still disappointed by the small number of experiments**
> > >
> > > When conducting experiments we were doing experiments to test a particular hypothesis:
> > >
> > > - For random directions, we test with Mnist and copy task, and the idea was to test if the performance of proposed estimator improves when increasing the number of directions.
> > >
> > > - For Syngrad experiments, we test with copy task to check if the proposed estimator improves over syngrad baseline.
> > >
> > > - For meta-learning experiments, generally in meta-learning, we require to backprop through the inner loop, and hence we test the proposed estimator as compared to Reptile baseline by increasing the number of steps in the inner loop.
> > >
> > > - For truncated backprop experiment, we wanted to test if the dodge estimator can improve for shorter truncation lengths, as compared to truncated backprop.
> > >
> > > **new experiment result**
> > >
> > > We did another experiment on the character level language modelling where we used the candidate direction as a result of randomized truncation with max_trunc_len=3. We used a LSTM of 256 hidden units, and we report the test bpc (lower is better for DODGE, and the baseline). DODGE gets bpc of 1.51 whereas  the baseline bpc of 1.56. We note that we have not done exhaustive experiments by varying hyper-parameters (either for baseline or for proposed method) as well as by varying the max_trunc_len. Preliminary evidence suggests that DODGE improves over truncated backprop for shorter truncation length (3 in this case).
> > >
> > > We note that we evaluate the proposed idea using different candidate directions (random directions, directions as a result of truncated backprop, directions as a result of synthetic gradient, and reptile) on different tasks such as copy task, NeRF task and the MNIST task. On the MNIST task, we also showed that the performance of the proposed method can be improved on using multiple candidate directions. In general, more experiments can always be done to test the proposed method, but given the number of different candidate directions, and different tasks, we feel like we have already shown the breadth and width of the utility of the proposed method.

---

### Official Review · Reviewer_GYWv · 2021-11-04

**Correctness:** 3
**Technical Novelty And Significance:** 1
**Empirical Novelty And Significance:** 3
**Recommendation:** 5
**Confidence:** 3

**Main Review:**

The idea is simple and intuitive: Any vector $u$ can be turned into a
descent direction by computing
$\langle \nabla L(w), u \rangle u$, where the inner product may be obtained at a lower cost than computing
the gradient $\nabla L(w)$ itself.

This idea has been known in the optimization literature as a "sketch and
project" gradient estimate (see, e.g., [1] and citations therein).
This line of work should be cited and duly acknowledged.
To my knowledge, it hasn't been studied specifically in the context of
recurrent ML architectures, so a paper like the present one could still have its merits.

The computational cost of the proposed approach is not discussed in
sufficient detail.
While the memory savings compared to (T)BPTT
are apparent, I don't see a reason why the *computational* complexity would
be any lower.
We are still relying on a gradient computation of $\frac{dx_t}{dw}$, even if
we immediately project that partial derivative onto the candidate direction.
Unless I am missing something, this cost is *added* to the cost of computing
the candidate direction.
That means that using directional gradient trick in conjunction with a candidate
direction based on TBPTT/DPG/SYN would (at least) double the computational
cost. Could the authors please clarify this aspect?

The experimental comparison is a bit lacking in my opinion.
On every problem, only one competing method is compared to. Why not report
the performance of all baseline methods for each test problem?

Except for the lacking discussion of computational cost, the paper is clear and well-written.


[1] https://proceedings.neurips.cc/paper/2018/file/fc2c7c47b918d0c2d792a719dfb602ef-Paper.pdf

**Summary Of The Paper:**

The paper proposes directional gradient descent as a way to approximate
the gradient in recurrent learning. Directional gradient descent
uses a candidate direction (e.g., a random vector or a biased gradient
estimate) and then "corrects" that estimate by scaling it with the
directional derivative, i.e, the inner product between the candidate
direction and the true gradient (or an unbiased estimate thereof).

**Summary Of The Review:**

Essentially, the paper is applying the known "sketch and project" trick
to recurrent learning, so the novelty is rather limited. The experimental
comparison is a bit lacking in my opinion and there is a big question mark in terms of the added computational cost
(and whether that is worth it in practice). I am recommending rejection
for now, but would encourage the authors to respond to my concerns in the rebuttal.

---

> ### Author Response · Authors · 2021-11-18
> **Computational Cost and "sketch and project" related work**
>
> We thank the reviewer for taking time and providing feedback on the paper. We are enthused that the reviewer found the idea simple and intuitive.
>
> **This idea has been known in the optimization literature as a "sketch and project" gradient estimate (see, e.g., [1] and citations therein)**
>
> We thank the reviewer for pointing this line of research. We were indeed not aware of the “sketch and project” work. We would properly cite and discuss this line of work. As reviewer mentioned, we use the general “sketch and project” for training of the recurrent functions, and in the context of meta-learning. We also considered adaptive estimates of the true gradients (TBPTT, Syngrad, Reptile). We also note that on the MNIST task, we showed that the performance of the proposed method can be improved by using multiple candidate directions.
>
> **The computational cost of the proposed approach is not discussed in sufficient detail.**
>
> We thank the reviewer for pointing this out. We agree with the reviewer that it may be confusing to the reader: the computational cost of the proposed method is double than that of the forward computation of the function (one forward pass for computing the direction, and another forward pass for computing the directional derivative). We also note that the proposed method scales much better as compared to other methods in the literature such as real-time recurrent learning (RTRL) (Williams & Zipser, 1989).
>
> **The experimental comparison is a bit lacking in my opinion. On every problem, only one competing method is compared to. Why not report the performance of all baseline methods for each test problem**
>
> We note that we consider different tasks and different ways to compute the candidate direction such as randomly sampled directions, truncated back-propagation through time (BPTT), the “Reptile” meta-learning approach and synthetic gradients.
>
> We tried to compare the respective baselines for different tasks. For example. For the Nerf task, we compared to the reptile baseline (as reptile would be a natural baseline, UORO or Syngrad have not been used in such context before)  for the copy task, we compared to UORO, Syngrad, TBPTT. The goal of the  MNIST experiments was to show that the performance of the proposed method improves as we increase the number of directions.
>
> Are there any specific experiments which the reviewer may like to see ?

---

> > ### Author Response · Authors · 2021-11-24
> > **Further Clarifications ?**
> >
> > Dear. Reviewer,
> >
> > We have updated the paper discussing sketch and project literature and discussing computational details. We would like to thank the reviewer, as we believe it has improved the clarity as well as the presentation of the work.
> >
> > Would the reviewer has any updated impression of our work ?
> >
> > Thanks for your help and time.

---

> > > ### Comment · Reviewer_GYWv · 2021-11-25
> > > **Thanks for the revision**
> > >
> > > Thank you for updating the manuscript regarding the sketch-and-project reference. However, I don't think the added paragraph adequately positions the paper in the context of previous work. You write: "The proposed estimator is also related to the “sketch and project” gradient estimate." However, your proposed method **is** a sketch-and-project gradient estimate. So I would say that your paper studies/applies the sketch-and-project technique in the context of dynamical methods. Also the paper by Hanzely et al. (2018) is not necessarily the best general reference for sketch-and-project, it was just the first paper that popped into my head...
> > >
> > > The discussion of the computational cost has improved a lot, but I think the paper could still be clearer, e.g., it should be made clear that the directional derivative has cost $O(p)$ per candidate direction, so the variants involving multiple candidate directions are more costly.

---

> > > > ### Author Response · Authors · 2021-11-25
> > > > **Thanks for your reply.**
> > > >
> > > > We appreciate the reviewer's reply.
> > > >
> > > > Since we can't update the paper now, we would be happy to update the relevant section like this:
> > > >
> > > > "The proposed estimator is an instantiation  of the “sketch and project” gradient estimate (Hanzely et al., 2018). These methods uses the information provided in the sketch to construct an unbiased estimator of the gradient. **Here, we consider adaptive estimates of the true gradient to provide a candidate direction (i.e., to construct the sketch)**. To the best of our knowledge, these methods have not been studied specifically in the context of dynamic models (i.e., training of the recurrent architectures)
> > > >
> > > > **I think the paper could still be clearer**
> > > >
> > > > We would update it like this:
> > > >
> > > > "For a single candidate direction, the computational cost of the proposed method is double than that of the forward computation of the function i.e., one forward pass for computing the candidate direction, and another forward pass for computing the directional derivative. We note that the complexity scales while using multiple candidate directions as the computational cost for directional derivative is O(p) per candidate direction."
> > > >
> > > > We will update the paper in the next revision, and also happy to upload the updated paper anonymously (if it's allowed and would be helpful for the reviewer).
> > > >
> > > > Does this answer reviewer's concerns ?

---

> > > > > ### Author Response · Authors · 2021-11-29
> > > > > **Further Clarifications**
> > > > >
> > > > > Dear. reviewer,
> > > > >
> > > > > Thanks again for engaging, and providing useful feedback. The feedback from the reviewer has helped to contextualize the work in the relevant literature (which we were not aware of), and have improved the presentation of the paper.
> > > > >
> > > > >  Since the review period is coming to an end, we would like to know if the above reply would satisfy reviewer's concern.
> > > > >
> > > > > Thanks for your help and time.

---

### Official Review · Reviewer_gsh8 · 2021-11-08

**Correctness:** 3
**Technical Novelty And Significance:** 3
**Empirical Novelty And Significance:** 3
**Recommendation:** 6
**Confidence:** 3

**Main Review:**

Strengths:

- The paper is working on a new approach to gradient-based optimization for recurrent architectures and doing this with an approach that is refreshing to see.

- The paper has empirical evidence that the proposed method works across a range of gradient estimators from BPTT, Reptile etc.

Weaknesses:

- One of the main concerns I have with the paper is the assertion that computing the directional derivative has “computational cost the same as the forward computation of the function”, which the authors mention several times in the paper, referring to the cost of computing the directional derivative with forward mode automatic differentiation. I believe this doesn’t cover the full picture and can be misunderstood by the reader.

I think what the authors mean by this statement is what they describe in section 3.2: that the directional derivative can be computed with cost of O(p) per time step where p is the size of the parameter vector with respect to which the gradient is computed, and this has the same computational complexity with the forward computation of the original function (without the derivative). This might be correct, but it doesn’t mean that the directional derivative can be computed for free with a “computational cost the same as the forward computation of the function” which to me seems to imply that both the function and the directional derivative can be computed without any extra computational cost compared with just the forward computation of the original function (without the derivative). In other words, I believe that when computing the directional derivative and the original function together, as it happens during forward mode automatic differentiation, the cost is indeed more (up to double) compared with the computation of the original function (without the derivative). In any case, it would be good if the authors clarify these points in the paper overall.

- In Section 2, it would really help to have a figure explaining the dependencies between x_t, l_t, etc. in the BPTT and synthetic gradient explanations in equations 1 and 2.

- In Section 2, synthetic gradients, can you give a bit more detail about how the bootstrapped terms explained immediately after equation 2 are used in practice in an implementation?

- In Section 2, real-time recurrent learning, the notation of z = h_T(...(h_1 (y)) … ) is slightly out of place given the use of x, l, w, etc. up to that point.


**Summary Of The Paper:**

The paper proposes a new gradient-based learning algorithm for recurrent neural networks, making use of the directional derivative along a candidate direction. The directional derivative serves the purpose of improving the usefulness of a given candidate direction for gradient-based parameter updates by computing the projection of the gradient along the given direction. The candidate direction can come from various sources such as randomly sampled directions, truncated backpropagation through time (BPTT), the “Reptile” meta-learning approach by Nichol et al. (2018), or synthetic gradients. The authors call this technique “deep online directional gradient estimate” (DODGE) and demonstrate it in several experiments including a copy task (Graves et al. 2014) and a NeRF task (Mildenhall et al. 2020).

**Summary Of The Review:**

I have a positive view of this paper and the line of research the authors are working on. I believe there is a lot of value in revisiting earlier work such as real-time recurrent learning (RTRL) (Williams & Zipser, 1989), which the authors correctly identify as a related work and explain in their manuscript.

---

> ### Author Response · Authors · 2021-11-18
> **Computational complexity of the proposed method.**
>
> We thank the reviewer for taking time and providing feedback on the paper. We believe the reviewer’s comments have helped to improve the presentation of the paper.
>
> **One of the main concerns I have with the paper is the assertion that computing the directional derivative has “computational cost the same as the forward computation of the function**
>
> We thank the reviewer for pointing this out. We agree with the reviewer that it may be confusing to the reader: the computational cost of the proposed method is double than that of the forward computation of the function (one forward pass for computing the direction, and another forward pass for computing the directional derivative. We also note that the proposed method scales much better as compared to other methods in the literature such as real-time recurrent learning (RTRL) (Williams & Zipser, 1989).

---

### Author Response · Authors · 2021-11-20
**Updated Manuscript**

Dear Reviewers,

We have updated the manuscript referencing "sketch and project" idea in the optimization literature and discussing the computational complexity of the proposed method.

Thanks for your time, and useful feedback.

---

### Public Comment · ~Nick_McGreivy1 · 2022-02-23
**Nice paper, two possible concerns**

1. I like figure 1, but when I tried to reproduce this, I found D=1 to be unstable. Can you run for, say, 100,000 iterations and see if D=1 goes unstable?

2. My intuition for this method (gained from MNIST and figure 1) is that even at low D DODGE gives relatively low values of the loss but performs poorly on other evaluation metrics, such as training accuracy or perhaps PSNR. Based on this intuition, we would expect DODGE to perform reasonably well in comparison to BPTT in figure 2 and figure 4, which it does. Weirdly, figure 3 compares DODGE and reptile on PSNR, but does not compare to BPTT or TBPTT. Is it possible to do that comparison?

---

### Decision · Program_Chairs · 2022-01-20

**Decision:**

Accept (Poster)

**Comment:**

This paper adapts a method called "real-time recurrent learning" for training recurrent neural networks. The idea is to project the true gradient onto a subspace of desired dimensionality along a candidate direction. There are a variety of possible candidates: random directions, backpropagation through time, meta-learning approaches, etc.

The main strength of the paper is that it is a very simple idea that seems to have practical utility.

While often presented in different contexts, it should be clearly noted by the authors that the general idea of using low dimensional directional derivatives for computational efficiency is fairly common in optimization. Reviewers mention sketch and project methods. This has also been looked, for example, in the context of Bayesian optimization, with [random selection](https://bayesopt.github.io/papers/2016/Ahmed.pdf) and [value of information based](https://proceedings.neurips.cc/paper/2017/file/64a08e5f1e6c39faeb90108c430eb120-Paper.pdf) criteria.


Reviewers appreciated aspects of the paper, though had concerns about relations to sketch and project methods, computational costs, and experimental demonstrations and baselines. Through the rebuttal period, reviewers were mostly satisfied that the concerns about computational costs were well-addressed. A better job could still be done about describing relation to other work. There was also still some desire for more thorough experimental demonstrations and consistent baselines, as described in the reviews. The paper also could use some additional proof-reading as it contains several grammatical errors. On the whole, the paper makes a nice simple practical contribution. Please carefully account for reviewer comments in updated versions.